**Data Availability Statement:** All relevant data are within the manuscript and its Supporting information files.

# Clinical and biological correlates of morning serum cortisol in children and adolescents with overweight and obesity

Anton Martens[1], Bünyamin Duran[2], Jesse Vanbesien[1], Stephanie Verheyden[1], Bart Rutteman[1], Willem Staels[1,3], Ellen Anckaert[4], Inge Gies[1,5], Jean De Schepper[1,6]*

1 Division of Pediatric Endocrinology, Department of Pediatrics, Universitair Ziekenhuis Brussel, Vrije Universiteit Brussel, Brussels, Belgium, 2 Faculty of Medicine and Pharmacy, Vrije Universiteit Brussel, Brussels, Belgium, 3 Research group BENE, Vrije Universiteit Brussel, Brussels, Belgium, 4 Department of Clinical Chemistry, Universitair Ziekenhuis Brussel, Vrije Universiteit Brussel, Brussels, Belgium, 5 Research group GRON, Vrije Universiteit Brussel, Brussels, Belgium, 6 Research group BITE, Vrije Universiteit Brussel, Brussels, Belgium

* jean.deschepper@uzbrussel.be

## Abstract

### Background and aim

A fraction of children with obesity have increased serum cortisol levels. In this study, we describe the clinical characteristics of obese children and adolescents with elevated morning serum cortisol levels and the relationship between the cortisol levels and components of the metabolic syndrome.

### Methods

Retrospective medical record review study of children aged 4 to 18 years with overweight or obesity seen for obesity management in the Pediatric Obesity Clinic of the UZ Brussel between 2013 and 2015.

### Results

A total of 234 children (99 boys and 135 girls) with overweight (BMI z-score > 1.3) without underlying endocrine or genetic conditions were included. Mean (SD) age was 10.1 (2.8) years, BMI SD-score 2.5 (0.6), and body fat percentage 37% (7.9). Serum fasting cortisol levels were elevated (>180 µg/L) in 49 children, normal (62–180 µg/L) in 168, and decreased (<62 µg/L) in 12. Serum fasting cortisol was not significantly correlated with gender, age, or degree of adiposity. But correlated significantly with fasting glucose ($R_s = 0.193$; $p < 0.005$), triglycerides ($R_s = 0.143$; $p < 0.05$), fibrinogen ($R_s = 0.144$; $p < 0.05$) and leptin levels ($R_s = 0.145$; $p < 0.05$). After adjustment for serum insulin and leptin, the correlation between serum cortisol and fasting glucose remained significant.

### Conclusion

Elevated morning serum cortisol levels were found in 20% of overweight or obese children and adolescents, irrespective of the degree of adiposity, and were associated with higher

**Funding:** This work was supported by the Universitaire Stichting van België (WA-0337). The funder had no role in study design, data collection and analysis, decision to publish, or preparation of the manuscript. WS holds a senior clinical investigator grant from the Research Foundation Flanders (File number: 77833).

**Competing interests:** The authors have declared that no competing interests exist.

fasting glucose, irrespective of underlying insulin resistance. The long-term cardiometabolic consequences of hypercortisolemia in childhood obesity needs further study.

## Introduction

Approximately 10 to 15% of Belgian youth are overweight and nearly 5% are obese [1]. Primary obesity in childhood is associated with several metabolic changes, such as high triglycerides and high fibrinogen levels, and endocrine changes, such as insulin resistance, high leptin, high adrenal androgens, and high cortisol levels [2,3]. Obese children are reported to have higher morning serum cortisol, reflecting the cortisol awakening response, as well as increased scalp hair cortisol, a marker for long-term cortisol exposure [4–11]. These increased serum cortisol levels have been related to overactivation of the hypothalamic-pituitary-adrenal (HPA) axis [12]. However, decreased serum cortisol levels secondary to an increased cortisol metabolic rate have also been reported [13,14]. While the evidence for systemic hypercortisolism in obesity is not conclusive, it is well established that present chronic cortisol excess causes multiple physiological changes: increased insulin resistance (by interfering with insulin signaling [15]), higher fasting glucose levels (by decreasing insulin action and glucose effectiveness [16]), higher blood pressure readings (by activating the mineralocorticoid receptor and increasing the vascular responsiveness [17]), and more abdominal fat accumulation (by promoting adipocyte differentiation through glucocorticoid receptor signaling, which are more abundant in visceral compared to subcutaneous fat [18]).

The consequences of obesity-associated hypercortisolism in obese children are understudied. Previous work reporting on serum morning cortisol was either restricted to Latino youth under 14 years of age [6], obese infants [7], or excluded subjects with arterial hypertension [8]. Most of the studies lacked data on body composition [4–9]. In this study, we report on the association between serum cortisol and components of the metabolic syndrome in obese children and adolescents between 4 and 18 years of age. Our aim was to clinically characterize overweight or obese youth with high morning cortisol and to study whether these children and adolescents are at higher risk for metabolic syndrome conditions such as high fasting glucose or triglycerides, insulin resistance, low HDL cholesterol, or high blood pressure compared to overweight or obese youth with normal cortisol levels.

## Methods and patients

In this retrospective medical record review study, we included youth between 4 and 18 years of age who were referred to the Pediatric Obesity Clinic of the UZ Brussel between 2013 and 2015 for obesity management. Medical records were reviewed by trained physicians and used to retrieve metabolic, hormonal, and body composition results. Only children with a BMI SDS >1.3 were included. Females taking oral contraceptives and children with secondary obesity were excluded. Before starting multidisciplinary obesity therapy, routine clinical and biological evaluations were done to exclude secondary obesity, including hypothyroidism and hypercortisolism, and to evaluate the presence of metabolic complications, including dyslipidemia, liver steatosis, insulin resistance, arterial hypertension, and low-grade inflammation. Standard anthropometry, blood pressure measurement, body composition analysis by bioelectrical analysis, and blood analysis were performed in fasting conditions. The patients and their parents provided written informed consent, and the study was conducted following institutional

guidelines. The study was approved by the UZ Brussel ethical committee (File 2018/140, B.U.N. 143201835934).

## Anthropometry and body composition

Standing height was measured using a stadiometer (Seca 217, Hamburg, Germany) and body weight was measured using an electronic scale (Seca 877 type 3 scale, Hamburg, Germany). Waist circumference was taken midway between the lowest ribs and the iliac crest at the end of a normal expiration with a Lufkin steel anthropometric tape (Lufkin W606PM). Body composition was assessed by bio-electrical impedance analysis using the Bodystat 1500 (Bodystat, United Kingdom). Blood pressure was taken after a 5-minute rest, using an oscillometric Mindray VS-900 (Mindray, Shenzen, P.R. China).

## Blood sample collection and analysis

The blood samples were collected between 7 and 10 a.m. after an overnight fast. Glucose was measured in lithium-heparin (LiHep) plasma using the hexokinase assay on a Cobas 8000 C702 / Roche Diagnostics platform. Total cholesterol, LDL-cholesterol, and HDL-cholesterol were measured in LiHep plasma using the cholesterol oxidase/peroxidase assay (Cobas 8000 C702). Triglycerides were measured in LiHep plasma using the glycerol kinase/glycerol phosphate oxidase/peroxidase assay (Cobas 8000 C702). Fibrinogen was measured in sodium-citrate plasma using the turbidimetric clotdetection assay on an ACL Top Family / Instrumentation Laboratory platform. Hormone levels were measured using commercial immunoassays. Insulin was measured by electrochemiluminescence immunoassay (Cobas 8000 e801), leptin by ELISA (Wallace Victor2 TRF counter, Perkin Elmer), and cortisol by electrochemiluminescence immunoassay (Cobas 8000 e801). The intra-assay coefficient of variability of the cortisol assay is 2% at 121 μg/L and 2.1% at 272 μg/L. The upper limit for morning cortisol is 180 μg/L and the lower limit is 62 μg/L. All analyses were performed at UZ Brussel.

## Calculations and definitions

Body mass index (BMI) was calculated by dividing the weight in kilograms by height in meters squared ($kg/m^2$). Anthropometric values were expressed as standard deviation scores (SDS) or z-scores [z-score = (value − mean)/SDS] for gender and age according to national reference data [19]. Blood pressure values were expressed as SDS for gender, age, and height according to international reference data [20]. HOMA-IR was calculated according to the formula: fasting insulin (pmol/L) x (fasting glucose (mg/dL) / 2430. LDL cholesterol was calculated using the formula: total cholesterol − HDL cholesterol − (triglycerides / 5).

Overweight was defined by a BMI z-score > 1.3 and obesity by a BMI z-score > 2. An elevated fasting glucose concentration was defined by a fasting glucose above 100 mg/dl. Dyslipidemia was defined by a HDL cholesterol < 35 mg/dl, triglycerides > 150 mg/dl or a LDL cholesterol > 135 mg/dl. Elevated blood pressure was defined by either a systolic or diastolic blood pressure SDS > 2. Insulin resistance was defined by a HOMA-IR > 4. An elevated morning serum cortisol concentration was defined by a value > 180 μg/L and a decreased concentration by a value below 62 μg/L, while an elevated fibrinogen was defined by a value > 400 mg/dl.

## Statistical analysis

Descriptive statistics (mean, SD, median, and ranges) were used to describe the population characteristics. The Kruskal-Wallis test, followed by a Mann-Whitney U test, Student's t-test,

or Chi square test were used to compare differences between groups, as appropriate. Correlations between parameters were analyzed using Spearman rank correlation coefficients. The level of statistical significance was set at $P \leq 0.05$. Statistical analysis, including sample size calculation was conducted using IBM SPSS for Windows v27. Using a 2-sided test, 5% significance level test ($\alpha = 0.05$) with power 80% power ($\beta = 0.2$), the required sample size is approximate 198 to detect Spearman Rank order of 0.2 between cortisol and glucose.

## Results

### Clinical and biological characteristics

A total of 234 children with overweight (BMI z-score > 1.3) without underlying endocrine or genetic conditions were included in this study. Clinical characteristics (mean (SD)) are presented in Table 1. Their age ranged between 4 and 17.4 years, BMI z-score between 1.4 and 5.72 and body fat % ranged between 17 and 69%. Of the children with a complete medical record, 11 (5%) had a birth weight SDS < 2, 35 (15%) patients had a BMI SDS > 3 and 15 (7%) had a body fat percentage >50%. Of the screened patients 31 (15%) had a systolic blood pressure SDS >2, 90 (40%) had a HOMA-IR >4, while 6 (2.6%) had abnormal fasting glucose levels (between 100 mg/dl and 126 mg/dl), 27 (12%) had an elevated fibrinogen (> 400 mg/dl), 30 (13%) had elevated triglycerides (> 150 mg/dl), 20 (9%) had an elevated LDL cholesterol (> 135 mg/dl), and 37 (16%) had decreased HDL cholesterol concentrations (< 35 mg/dl).

### Comparison between males and females

In total 99 male and 135 female overweight or obese children and adolescents were included. No significant gender-related differences were found in anthropometric data and body fat percentage (Table 1). Mean leptin levels were higher in girls than in boys (37 vs 29; $P < 0.001$). Mean fasting glucose was higher in boys (88 vs 86mg/dL; $P = 0.070$). Mean HDL was higher in boys (46 vs 43 mg/dL; $P = 0.035$). Serum cortisol levels were similar in both sexes (Table 2).

### Characterization of overweight children and adolescents with an elevated morning serum cortisol

A total of 49 children (20.9%, 20 male and 29 female) had elevated morning serum cortisol levels, while 12 (5.1%, 4 boys and 8 girls), had decreased cortisol levels. Age, birth weight SDS, BMI SDS, waist SDS, systolic (SBP) and diastolic (DBP) blood pressure SDS, and body fat (%) were not significantly different between the 3 groups (Table 3).

Of the studied analytes, glucose, fibrinogen and leptin differed significantly between the 3 cortisol subgroups. Subgroup analysis revealed that children with high morning serum cortisol levels (> 180 µg/L) had significantly higher fasting glucose (88 vs 84 mg/dL; $P < 0.05$) and higher fibrinogen levels (329 vs 275 mg/dL; $P < 0.5$) compared to those with low morning serum cortisol values (<62 µg/L) (Table 4). However, no significant differences were found between children with high (> 180 µg/L) and those with normal (62–180 µg/L) morning cortisol concentrations (Table 4).

### Correlates of serum cortisol with clinical characteristics and auxological measurements

Children under the age of 10 years had similar cortisol levels compared to older children. Serum cortisol did not correlate significantly with the other studied clinical characteristics such as age at presentation ($R_s = 0.114$; $P = 0.081$) and BIA body fat percentage ($R_s = 0.043$; $P = 0.547$) or auxological parameters including birth weight SDS ($R_s = -0.077$; $P = 0.265$),

**Table 1. Clinical characteristics and comparison between boys and girls.**

| Parameters | Total population (N = 234) | Boys (N = 99) | Girls (N = 135) | P value |
|---|---|---|---|---|
| | Mean (SD) | | | |
| Age (years) | 10.1 (2.9) | 10.0 (2.8) | 10.1 (2.9) | 0.803 |
| Gestational age (weeks) | 39.6 (1.9) | 39.2 (1.7) | 39.2 (2.0) | 0.885 |
| Birth weight (g) | 3388 (618) | 3446 (630) | 3345 (607) | 0.230 |
| Birth length (cm) | 50.0 (2.6) | 50.4 (2.7) | 49.7 (2.4) | 0.070 |
| Weight (kg) | 61.1 (24.7) | 61.3 (24.7) | 61.0 (24.8) | 0.942 |
| Length (cm) | 145.1 (15.7) | 145.8 (16.0) | 144.5 (15.5) | 0.528 |
| BMI (kg/m$^2$) | 28.1 (7.1) | 27.4 (5.8) | 28.5 (7.8) | 0.240 |
| BMI SDS | 2.45 (0.58) | 2.41 (0.48) | 2.48 (0.64) | 0.322 |
| Waist circumference (cm) | 1.8 (13.6) | 82.8 (14.4) | 81.0 (12.9) | 0.322 |
| Waist circumference SDS | 2.26 (0.61) | 2.25 (0.53) | 2.27 (0.67) | 0.812 |
| BIA body fat (%) | 37.2 (7.9) | 36.2 (8.2) | 37.9 (7.6) | 0.139 |

weight SDS ($R_s$ = 0.005; P = 0.944), BMI SDS ($R_s$ = 0.023; P = 0.723), or waist SDS ($R_s$ = 0.062; P = 0.371) (Table 5).

## Correlates of serum cortisol with cardiometabolic parameters

Serum morning cortisol was found to correlate significantly with fasting glucose ($R_s$ = 0.193; P < 0.05), triglycerides ($R_s$ = 0. 143; P < 0.05), fibrinogen ($R_s$ = 0.144; P < 0.05), and leptin ($R_s$ = 0.145; P < 0.05). Of note, adjusting serum cortisol for concurrent insulin and leptin concentrations, did not change the correlation between serum cortisol and fasting glucose. Serum morning cortisol did not correlate significantly with DBP SDS ($R_s$ = 0.081; P = 0.252), SBP SDS ($R_s$ = 0.127; P = 0.071), HDL cholesterol ($R_s$ = 0.085; P = 0.198), LDL cholesterol ($R_s$ = 0.051; P = 0.439), fasting insulin ($R_s$ = 0.078; P = 0.245), and HOMA-IR ($R_s$ = 0.098; P = 0.141) (Table 6).

## Discussion

Elevated morning cortisol levels were found in one-fifth of a large population of overweight and obese Belgian children and adolescents, seeking medical treatment for obesity. Morning cortisolemia was found to be independent of gender, age, and measurements of body fat,

**Table 2. Biological characteristics and comparison between boys and girls.**

| Parameters | Total population (N = 234) | Boys (N = 99) | Girls (N = 135) | P value |
|---|---|---|---|---|
| | Mean (SD) | | | |
| Glucose (mg/dL) | 86 (6) | 88 (6) | 86 (7) | 0.070 |
| Insulin (pmol/L) | 135 (82) | 133 (93) | 136 (73) | 0.815 |
| HOMA-IR | 4.1 (3) | 4.1 (3) | 4.1 (2) | 0.928 |
| Cholesterol (mg/dL) | 165 (29) | 168 (29) | 163 (29) | 0.197 |
| HDL-cholesterol (mg/dL) | 45 (11) | 46 (10) | 43 (11) | **0.035** |
| LDL-cholesterol (mg/dL) | 99 (26) | 101 (25) | 98 (26) | 0.389 |
| Triglycerides (mg/dL) | 104 (55) | 101 (57) | 106 (53) | 0.519 |
| Fibrinogen (mg/L) | 319 (65) | 310 (68) | 325 (61) | 0.069 |
| Leptin (µg/L) | 33 (18) | 29 (16) | 37 (19) | **< 0.001** |
| Cortisol (µg/L) | 138 (58) | 139 (55) | 137 (60) | 0.754 |

**Table 3. Comparison of clinical characteristics between overweight/obese children with low, normal, and high cortisol.**

| Parameters | Low cortisol (< 62 µg/L) | | Normal cortisol (62–180 µg/L) | | High cortisol (>180 µg/L) | | P value |
|---|---|---|---|---|---|---|---|
| | | | *Median (range)* | | | | |
| | | N | | N | | N | |
| Age (years) | 7.9 (6.1–13.8) | 12 | 9.9 (4.0–16.7) | 173 | 10.3 (6.2–17.4) | 49 | 0.530 |
| Birthweight (SDS) | 0.35 (-1.60–2.03) | 8 | 0.03 (-3.04–3.44) | 160 | -0.21 (-2.18–2.27) | 46 | 0.725 |
| BMI (SDS) | 2.43 (1.70–3.10) | 12 | 2.42 (1.40–4.32) | 173 | 2.35 (1.61–5.72) | 49 | 0.853 |
| Waist circumference (SDS) | 2.18 (1.59–2.81) | 10 | 2.24 (0.88–4.14) | 156 | 2.30 (0.44–5.54) | 47 | 0.858 |
| SBP (SDS) | 0.55 (-2.29–3.36) | 12 | 0.78 (-3.09–3.92) | 147 | 0.99 (-2.06–3.84) | 44 | 0.397 |
| DBP (SDS) | 0.58 (-1.22–1.89) | 12 | 0.71 (-1.93–3.42) | 147 | 0.72 (-0.84–2.85) | 44 | 0.792 |
| BIA body fat (%) | 36.6 (30.0–41.7) | 10 | 35.2 (24.0–61.7) | 145 | 35.5 (24.0–69.4) | 43 | 0.952 |

whereas it correlated positively with fasting glucose, irrespective of fasting insulin. We could not differentiate obese children and adolescents with high cortisol from those with normal cortisol using clinical or biological characteristics. While in some studies higher morning serum cortisol concentrations have been found in obese children [4–10], normal and lower morning serum and salivary concentrations have been reported by others [14,21–23]. As the cortisol secretion is both diurnal and pulsatile, conflicting results concerning the association between cortisol secretion and obesity can be expected by using single-point sampling. Besides, sampling at different moments in the morning (between 08 a.m. and 10 a.m.) and the use of different analysis techniques (immuno-assay versus LC-MS/MS) might further contribute to the variation in serum or salivary cortisol concentrations. Moreover, some characteristics that are innate to the study population are known to influence serum cortisol concentrations (a history of intrauterine growth restriction, the duration of obesity, obesity-associated metabolic disturbances such as insulin resistance, sleep disturbances, emotional and/or behavior problems) and may explain the discrepancies in morning cortisol levels observed in different studies. The study population in this report had a moderate degree of obesity (mean BMI SDS of 2.45, mean body fat percentage BIA of 37%, and mean HOMA-IR of 4.1). Intrauterine growth restriction, known to increase serum cortisol and fasting glucose in childhood, was only present in 5% of the studied subjects and did not correlate with hypercortisolemia [24]. We were unable to study the impact of the duration of obesity, the presence of obstructive sleep apnea syndrome or sleep deprivation, and the emotional status of the included subjects due to incomplete data. Anxiety, depression, and chronic stress may all contribute to an activated

**Table 4. Comparison of biological characteristics between overweight/obese children with low, normal, and high cortisol.**

| Parameters | Low cortisol (< 62 µg/L) | | Normal cortisol (62–180 µg/L) | | High cortisol (>180 µg/L) | | P value |
|---|---|---|---|---|---|---|---|
| | | | *Median (range)* | | | | |
| | | N | | N | | N | |
| Glucose (mg/dL) | 84 (77–92) | 12 | 87 (71–112) | 168 | 88 (74–105) | 49 | **0.050** |
| Insulin (pmol/L) | 98 (43–190.3) | 12 | 133 (21–495) | 166 | 151 (42–414) | 48 | 0.141 |
| HOMA-IR | 2.9 (1.2–6.2) | 12 | 4.0 (0.6–16.0) | 166 | 4.6 (1.4–12.5) | 48 | 0.091 |
| Cholesterol (mg/dL) | 166 (111–240) | 12 | 163 (76–275) | 168 | 168 (108–227) | 49 | 0.522 |
| HDL-cholesterol (mg/dL) | 46 (35–60) | 12 | 44 (17–82) | 168 | 46 (23–72) | 49 | 0.571 |
| LDL-cholesterol (mg/dL) | 103 (75–166) | 12 | 99 (40–174) | 168 | 100 (46–152) | 49 | 0.865 |
| Triglycerides (mg/dL) | 89 (39–178) | 12 | 102 (36–356) | 168 | 113 (46–342) | 49 | 0.087 |
| Fibrinogen (mg/L) | 275 (205–393) | 12 | 319 (123–524) | 164 | 329 (209–514) | 49 | **0.026** |
| Leptin (µg/L) | 20 (9–33) | 12 | 34 (6–145) | 173 | 36 (7–99) | 49 | **0.011** |

**Table 5. Correlation ($R_s$) between serum cortisol and auxologic parameters.**

| Parameter | Rs | P Value |
|---|---|---|
| Age | 0.114 | 0.081 |
| Birth weight SDS | -0.077 | 0.265 |
| Weight SDS | 0.005 | 0.944 |
| BMI SDS | 0.023 | 0.723 |
| Waist SDS | 0.062 | 0.371 |
| BIA body fat | 0.043 | 0.547 |

hypothalamic adrenal axis, reflected by increased morning cortisol concentrations. However, despite an increased prevalence of self-reported anxiety and depressive symptoms in obese children, morning cortisol concentrations did not differ between children with or without depression or anxiety [25–27]. Short duration of sleep is known to be associated with higher morning cortisol levels in healthy children, and both low and high morning salivary cortisol concentrations have been documented in children with obstructive sleep apnea. But the association between morning cortisol and sleep duration, sleep apnea frequency and tonsillar size has not been studied in obese populations [28–30].

Elevated morning cortisol concentrations have been reported in up to 14% of obese children and adolescents, mainly coinciding with insulin resistance [6,8–10]. However, the reliability of morning serum cortisol values is limited when used to assess hypercorticism. Most studies that used hair cortisol, a good measure of long-term (several months) exposure to circulating cortisol levels, found elevated cortisol concentrations in obese children [11,31,32].

Serum morning cortisol levels did not correlate with age, or gender in our sample. Similar findings were observed in previous studies of healthy children [33] and other studies of obese children and adolescents [8,34]. We found no association between classic anthropometric adiposity measurements and morning cortisol concentrations, confirming the results of other pediatric studies [8,34,35]. This is in contrast with some data in adolescent and adult populations [36–38]. This difference in the relationship between cortisol and visceral fat accumulation may relate to visceral obesity being a longer term consequence of prolonged glucocorticoid exposure up to adulthood [36,38,39]. However, other investigators found that serum morning cortisol and urinary cortisol excretion (both nighttime and 24h collections) correlated with the fat mass in the upper body regions, measured using whole-body DXA, in obese children and adolescents [36,40]. Moreover, in overweight Latina youth, serum cortisol

**Table 6. Correlation between serum cortisol and cardiometabolic parameters.**

| Parameter | $R_s$ | P Value |
|---|---|---|
| Glucose | 0.193 | **0.003** |
| Insulin | 0.078 | 0.245 |
| HOMA-IR | 0.098 | 0.141 |
| Cholesterol | 0.112 | 0.098 |
| HDL-cholesterol | 0.085 | 0.198 |
| LDL-cholesterol | 0.051 | 0.439 |
| Triglycerides | 0.143 | **0.030** |
| Fibrinogen | 0.144 | **0.031** |
| Leptin | 0.145 | **0.027** |
| SBP SDS | 0.127 | 0.071 |
| DBP SDS | 0.081 | 0.252 |

was found to correlate with intra-abdominal adipose tissue as assessed by magnetic resonance imaging, but not with waist circumference, revealing the inaptitude of this measurement in children and adolescents to reflect the visceral fat accumulation [6].

Of the investigated cardiometabolic risk factors, the most expected association was the correlation between morning cortisol and glucose levels, as cortisol is a glucocorticoid. This positive, albeit weak, correlation is in accordance with other studies in obese children and adolescents [4,6,8,35,38]. In our study, morning serum cortisol was determined at approximately 08 a.m. or one hour after wakening, reflecting most probably the diurnal peak in cortisol output. This diurnal cortisol rhythm, which is regulated by the central clock, has been held responsible for the circadian variation in plasma glucose, free fatty acids, and insulin [41]. The association between morning cortisol and glucose in our study might thus be explained by this circadian regulation and is also in line with the glucocorticoid effect on hepatic gluconeogenesis, which is maximal at awakening [42]. The association between serum cortisol and glucose persisted after correction for serum insulin, confirming that cortisol excess may impair glucose intolerance by decreasing not only the insulin action, but also glucose effectiveness [16,42]. On the other hand, we cannot exclude that the acute stress induced by taking blood might have elevated both serum glucose and cortisol.

Increased cortisol output in obesity has been linked to arterial hypertension through its effect on salt and water retention or vascular smooth muscle tone [17,43]. In several studies of obese children and adolescents, morning cortisol was weakly associated with SBP [4,6,8,10], but others found no significant differences in morning salivary concentrations in normotensive or hypertensive obese children [44]. We hypothesize that the narrow range of blood pressure, as well as the moderate degree and variation in adiposity, insulin resistance, and hyperleptinemia, might have obscured the correlation between morning cortisol and blood pressure readings in our study. Insulin resistance and hyperleptinemia have also been linked with arterial hypertension in obesity. As in other studies, fasting serum cortisol correlated with fasting leptin concentrations in our study [5,9]. The positive association between serum cortisol and leptin relates to the direct stimulatory effects of cortisol on leptin production [45,46]. In one study in obese youth a correlation between serum cortisol and fasting glucose was found in the absence of a correlation between cortisol and leptin, but the authors related the lack of the latter correlation to a limited sample size [34]. While the evidence is emerging that increased high-end cortisol contributes to increased cardiovascular risk by both local and systemic effects, few studies have investigated the relation between hypercortisolemia and vascular function [47,48]. Soriano-Rodríguez et al. found a significant correlation between serum morning cortisol and carotid intima thickness in prepubertal children [49].

The retrospective design, the use of a single morning cortisol measurement, and the lack of reliable quantification of abdominal fat are the most important limitations of this study. Selection bias may be present as the enrolled subjects are possibly highly motivated children or children in whom previous weight loss treatments have failed. Additionally, symptoms of stress or depression were not investigated, and blood pressure readings were only taken on a single occasion. This study has important strengths, including supplementary data on morning cortisol in a large group of obese children and adolescents, information on its relationship with adiposity, and several metabolic and endocrine measurements, allowing the adjustment for potentially confounding covariates.

In conclusion, elevated morning serum cortisol levels were present in 20% of overweight or obese children and adolescents. Obese youth with high serum cortisol could not be differentiated on clinical or biological grounds from those with a normal serum cortisol. Morning serum cortisol concentrations were found to correlate weakly with fasting glucose

concentrations, irrespective of insulin resistance. Further study exploring long-term cardiometabolic consequences of hypercortisolemia in childhood obesity is needed.

## Statement of ethics

The patients and their parents provided written informed consent for publication, and the study was conducted ethically in accordance with the World Medical Association Declaration of Helsinki and following institutional guidelines. The study was approved by the UZ Brussel ethical committee (File 2018/140, B.U.N. 143201835934).

## Supporting information

**S1 Dataset.**
(SAV)

## Author Contributions

**Conceptualization:** Anton Martens, Bünyamin Duran.

**Data curation:** Jean De Schepper.

**Formal analysis:** Jean De Schepper.

**Investigation:** Jean De Schepper.

**Methodology:** Willem Staels, Jean De Schepper.

**Project administration:** Anton Martens.

**Supervision:** Jesse Vanbesien, Stephanie Verheyden, Bart Rutteman, Willem Staels, Ellen Anckaert, Inge Gies, Jean De Schepper.

**Validation:** Willem Staels, Inge Gies, Jean De Schepper.

**Writing – original draft:** Anton Martens.

**Writing – review & editing:** Willem Staels, Inge Gies, Jean De Schepper.

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
