## [Decision Letter · Decision Letter 0]

18 Jun 2021

PONE-D-21-14498

Correlates of morning serum cortisol in children and adolescents with overweight and obesity

PLOS ONE

Dear Dr. Martens,

Thank you for submitting your manuscript to PLOS ONE. After careful consideration, we feel that it has merit but does not fully meet PLOS ONE’s publication criteria as it currently stands. Therefore, we invite you to submit a revised version of the manuscript that addresses the points raised during the review process.

We look forward to receiving your revised manuscript.

Kind regards,

Simone Perna, Ph.D

Academic Editor

PLOS ONE

Journal Requirements:

Reviewers' comments:

Reviewer's Responses to Questions

**Comments to the Author**

1. Is the manuscript technically sound, and do the data support the conclusions?

Reviewer #1: Partly

Reviewer #2: Yes

Reviewer #3: Yes

Reviewer #4: No

2. Has the statistical analysis been performed appropriately and rigorously? 

Reviewer #1: No

Reviewer #2: Yes

Reviewer #3: Yes

Reviewer #4: No

3. Have the authors made all data underlying the findings in their manuscript fully available?

Reviewer #1: Yes

Reviewer #2: Yes

Reviewer #3: Yes

Reviewer #4: No

4. Is the manuscript presented in an intelligible fashion and written in standard English?

Reviewer #1: No

Reviewer #2: Yes

Reviewer #3: Yes

Reviewer #4: No

5. Review Comments to the Author

Reviewer #1: Comments on “Correlates of morning serum cortisol in children and adolescents with overweight and obesity”

Summary and overall impression:

The study is a descriptive retrospective analysis studying fasting hyperglycemia, dyslipidemia, hyperfibrinogenemia as primary parameters/ or outcomes in overweight/obese children with high cortisol compared with those with normal cortisol level. However, more secondary parameters / or outcomes were mentioned later in the method, results, and discussion and were not referred to in the study aim and introduction.

Major changes:

1. Aim: To add an additional aim outlining the secondary cardio-metabolic parameters /outcomes that were assessed in the study.

2. Methods and patients: It will be better if you define the cutoff levels of cardio-metabolic parameters in the method and demonstrate the normal and high range. E.g., define hyperglycemia or dyslipidemia and so on. Also, there was not a cut-off level differentiating obese from overweight for further analysis.

3. Sample Size: How the sample size was tested and assured that it was large enough to generate reliable results?

Statistical Analysis:

The flow of the statistical analysis was towards proving the objective of the study, which is the correlation between elevated cortisol and hyperglycemia, dyslipidemia, hyperfibrinogenemia and hyperleptinemia.

Table 1: presented the anthropometric characteristics and compare them between boys and girls. The characteristics were stated in terms of the median and the range. The listed characteristics are usually normally distributed, and I suggest using mean ± sd to their measurements and parametric test for the comparisons.

Table 3 and table 5 presented the main outcomes of the study. In table 3 the biological characteristics were compared over the three subgroups of the population: low cortisol (<62 μg/L), normal cortisol (62-180 μg/L) and elevated cortisol (>180 μg/L). Through this classification it was recognized that the elevated cortisol subgroup represents about 20% of the total sample. The Kruskal Wallis p-values were not stated for significant comparisons between the three subgroups. The analysis was based on pairwise comparisons that identified the significance difference between the elevated cortisol subgroup from the lower cortisol subgroup with respect to fasting glucose, fibrinogen and leptin. The authors though did not mention the test on which the pairwise comparisons were made in their analysis on page 7.

The author has to take these points on board and rearrange the measured risk factors into primary and secondary in the aim and introduction sections, also rearrange results and tables display accordingly. Discussion has to go in systemically in the same flow of results display.

Reviewer #2: I am pleased to write my review on "Correlates of morning serum cortisol in children and adolescents with overweight and obesity" where authors investigated whether hypercortisolemia conferred an increased risk for fasting hyperglycemia, dyslipidemia, hyperfibrinogenemia, and hyperleptinemia in overweight children and adolescents. The article is well written and easy to understand but I would like to ask few concerns related to this study.

1. the size of population is not huge and why did the authors selected this particular time period for this study?

2. What do the authors think why morning cortisolemia was found to correlate positively with fasting glucose, irrespective of fasting insulin?

3. Have the authors check role of depression or anxiety in these participants for elevated morning cortisol concentrations?

4. Did the authors study role of sleep duration as it could be a reason for elevated morning cortisol level?

5. As children were in Pediatric Obesity Clinic for obesity management, can author comment on usage of any drugs or obesity management techniques which can result in stress?

Reviewer #3: The study is relevant and has scientific importance. Data were well analyzed. The results are similar to other published studies, which gives this manuscript importance. It is interesting that the authors continue to look for new answers to the initial questions.

Reviewer #4: General impression:

The study was based on retrospective medical record of children and adolescents with overweight or obesity problems seeking for obesity management. It was conducted to assess the relationship between hypercortisolemia and the risk of hyperglycemia, dyslipidemia, hyperfibrinogenemia, and hyperleptinemia in overweight children and adolescents. However, the MM, results and discussion cover diverse parameters, which were not specified precisely in the research objectives. The conclusion was not aligned with the main objective of the research.

Reviewing the manuscript revealed that the research idea has a potential for publication. However, setting objectives, MM, data analysis and presentation, depth of the discussion and engagement with published literatures and the conclusion all need rigorous amendments to achieve an acceptable level for publication. The current level of manuscript written language and flow of ideas hinders its suitability for publication. The extensive grammatical mistakes affected manuscript presentation in general causing it to be inappropriate for publication at this level.

Confidential detailed comments for the editors/authors:

Title: Needs modifications

I would suggest modifying the title to best reflect the research objective(s) and the measured parameters.

Abstract: Needs modifications

Along with the changes recommended in the various sections of the manuscript, you may notice that the abstract will demand subsequent amendments.

Introduction: Needs modifications

The introduction is very brief and does not cover all the measured substantial parameters. More emphasis should be done on reviewing published literature pertaining the relationship between hypercortisolemia and increased risk of hyperglycemia, dyslipidemia, hyperfibrinogenemia, and hyperleptinemia and the other measured parameters. You may like to elaborate on the effect of overweight/obesity on hypercortisolemia and how it affects the stated cardiometabolic parameters.

Engagement with published literatures can highlight what was/were the outcomes of the published research and hence you can emphasize the originality of your research.

The aim of the research must be supported with statements indicating the rationale behind selecting the studied parameters to assess the link between morning serum cortisol and hyperglycemia, dyslipidemia, hyperfibrinogenemia, and hyperleptinemia in overweight and obese patients. You may like to think of adding other objective(s) that can be related to the parameters analyzed in the study but not precisely stated in the objectives or focused on in the introduction.

Material and Methods (Methods and patients):

Needs modifications

The flow of this section is logic as it started with anthropometry and body composition analysis to obtain an overview of the studied sample. This was followed by blood sample collection and analysis, and finally the statistical analysis. However, this section needs to be revised. The following are some examples that need to be taken care of:

• Sample size: adequate sample size (n) should be calculated and specified precisely before proceeding with data analysis.

• The blood sample collection and analysis sections lacks rudimentary information that makes the work difficult to replicate and the results to be reproducible. E.g “Glucose, fibrinogen, total cholesterol, LDL-cholesterol, HDL- cholesterol, and triglycerides were measured by automated methods (Cobas 8000 C702)” it is not clear how samples were prepared for the automated analyzer and the analysis was performed. “Hormone levels were measured using commercial immunoassays”, it is not clear, which hormone they measured and what is the type and the manufacturer of the commercial immunoassays. More emphasis should be exerted on presenting details about the method of the analysis carried out. If details are not specified, then methodology should be supported with literatures.

• The collected data were subjected to Kruskal Wallace test, the Mann-Whitney U test, and the Chi square test to compare differences between groups. However, this is not always reflected in the results sections. E.g. Table 3 and the accompanied narrative.

• Clinical and biological characteristics indicates that the sample used in the study may be non-homogeneous, e.g. wide age groups, wide BMI and body fat % range. Did you perform homogeneity test between groups before performing the above analysis?

• What about the differences between overweight and obese patients? children and adolescents? can we consider them as one group?

Results: Needs modifications

In this part of the manuscript, you have presented several parameters and performed data analysis to achieve the main objective. Here are few comments that you may like to think about:

• The research depends on retrospective medical record of a wide age groups, which includes children and adolescents with diverse physiological status. Don’t you think, it would be more logic to divide the sample based on age groups with distinct physiological status?

• In the characterization of overweight children and adolescents with an elevated morning serum cortisol section and Table 3: the p-values need to be indicated to reveal the significant or non-significant differences between groups. I believe more emphasis should be attained on the difference between normal and elevated/reduced cortisol to meet the objective(s) of the study.

According to Table 3: are the results presented in this section related to obese patients only? Did you exclude overweight patients? Can you precisely identify the difference(s) between overweight and obese patients?

• In the correlates of serum cortisol with clinical characteristics and auxological measurements section: important data which reveals the differences between different pubertal stages are not shown, any reason for that? What is/are objective(s) of measuring the correlation between serum cortisol and clinical characteristics and auxological parameters? How this will help in achieving your objective.

• In the section which revels the correlates of serum cortisol with cardiometabolic parameters, what does the “r” value indicates? Do you think that there are strong relationships between the measured parameters? Refer to the conclusion.

• I think presenting data in the tables as means ± SD, is more meaningful compared to the median.

Discussion: Needs modifications

The discussion lacks deep and critical analysis of the research findings. You may like to provide strong and deep justification(s) to support your arguments in relation to the findings. Engagement with published literatures should be used to support your argument, it should not be presented as literature search to present general information.

Conclusion: Needs modifications

The conclusion is not satisfactory. It does not highlight the main outcome(s) of the research. You may like to link it with the objectives and the measured parameters to draw a clear and precise conclusion.

In addition, can the results presented in this study support your conclusion? i.e. “these increased levels correlated with higher fasting glucose concentrations” while r = 0.193?

References: Needs more engagement with relevant literatures

You needs to follow the referencing style recommended by the journal both in-test and in the list of references. Unified style is must.

6. PLOS authors have the option to publish the peer review history of their article (what does this mean?). If published, this will include your full peer review and any attached files.

Reviewer #1: No

Reviewer #2: **Yes: **Dr. Muhammad Nauman Zahid

Reviewer #3: No

Reviewer #4: No

---

## [Author Response · Author response to Decision Letter 0]

29 Jul 2021

This is the revised manuscript. The files now do meet the PLOS ONE's style requirements. In the first version, we included the phrase “data not shown” in the manuscript. As these date were not a core part of the research being presented in our study, we removed the phrase that refers to these data.

In the updated manuscript, we changed our title to better reflect the research objectives. We also updated our method section thoroughly, including the power calculations. Our tables were updated, and we present the relevant ones now in mean (SD). Extra relevant literature on mechanisms relating hypercortisolism and metabolic syndrome conditions was provided in the revised manuscript. 

For more in-depth information, we kindly refer to the 'response to reviewers' file.

---

## [Decision Letter · Decision Letter 1]

4 Oct 2021

Clinical and biological correlates of morning serum cortisol in children and adolescents with overweight and obesity

PONE-D-21-14498R1

Dear Dr. Martens,

We’re pleased to inform you that your manuscript has been judged scientifically suitable for publication and will be formally accepted for publication once it meets all outstanding technical requirements.

Kind regards,

Simone Perna, Ph.D

Academic Editor

PLOS ONE

Additional Editor Comments (optional):

Reviewers' comments:

Reviewer's Responses to Questions

**Comments to the Author**

1. If the authors have adequately addressed your comments raised in a previous round of review and you feel that this manuscript is now acceptable for publication, you may indicate that here to bypass the “Comments to the Author” section, enter your conflict of interest statement in the “Confidential to Editor” section, and submit your "Accept" recommendation.

Reviewer #1: All comments have been addressed

Reviewer #2: All comments have been addressed

Reviewer #3: All comments have been addressed

2. Is the manuscript technically sound, and do the data support the conclusions?

Reviewer #1: Yes

Reviewer #2: Yes

Reviewer #3: Yes

3. Has the statistical analysis been performed appropriately and rigorously? 

Reviewer #1: Yes

Reviewer #2: Yes

Reviewer #3: Yes

4. Have the authors made all data underlying the findings in their manuscript fully available?

Reviewer #1: Yes

Reviewer #2: Yes

Reviewer #3: Yes

5. Is the manuscript presented in an intelligible fashion and written in standard English?

Reviewer #1: Yes

Reviewer #2: Yes

Reviewer #3: Yes

6. Review Comments to the Author

Reviewer #1: Clinical and biological correlates of morning serum cortisol in children and adolescents with overweight and obesity

Summary and the overall:

Well written introduction, aim, methods, results, and discussion. Study limitations were well explained.

Major comments:

1-Aim: The aim of the study is now clearer and more inclusive. It covered both the primary and secondary parameters by using the term “components of the metabolic syndrome” which covers all the factors.

2- Methods & patients: All the parameters used in the study, were all defined. The cut-off levels were clearly noted.

3- Sample Size: The sample size calculation was conducted, and the required target was met. The power of 80% was met as well.

4- Discussion:

The discussion gave a clear clarification of the study limitations. One major limitation clearly demonstrated that different methods used for cortisol measurements has the potential to give different levels of cortisol. In addition, more limitations were related to data availability. Many factors that could be used to further correlate with high cortisol levels and obesity such as sleep obstructive apnea or emotional disturbances were missing in the available patient’s data used for this study.

The discussion was well written and organized. The major points of discussion were handled systematically and in the same flow of results displayed in the tables.

Reviewer #2: Authors have addressed my concerns and added limitations of their study, so I feel that article is good for publication now.

Reviewer #3: The authors answered all the questions asked by the reviewers. They added relevant information fact that contributed to a better understanding of the manuscript. I consider that the study is relevant, even in view of the limitations identified by the authors themselves, and should be accepted for publication.

7. PLOS authors have the option to publish the peer review history of their article (what does this mean?). If published, this will include your full peer review and any attached files.

Reviewer #1: No

Reviewer #2: **Yes: **Muhammad Nauman Zahid

Reviewer #3: No

---

## [Editor Report · Acceptance letter]

12 Oct 2021

PONE-D-21-14498R1 

Clinical and biological correlates of morning serum cortisol in children and adolescents with overweight and obesity 

Dear Dr. Martens:

I'm pleased to inform you that your manuscript has been deemed suitable for publication in PLOS ONE. Congratulations! Your manuscript is now with our production department. 

Kind regards, 

on behalf of

Professor Simone Perna 

Academic Editor

PLOS ONE